# A Low-Glycemic Index, High-Fiber, Pulse-Based Diet Improves Lipid Profile, but Does Not Affect Performance in Soccer Players

**DOI:** 10.3390/nu12051324

**Published:** 2020-05-06

**Authors:** Eliran Mizelman, Philip D. Chilibeck, Abdul Hanifi, Mojtaba Kaviani, Eric Brenna, Gordon A. Zello

**Affiliations:** 1College of Kinesiology, University of Saskatchewan, Saskatoon, SK S7N 5B2, Canada; Eli_Mizelman@sfu.ca (E.M.); mojtaba.kaviani@acadiau.ca (M.K.); eric.brenna@usask.ca (E.B.); 2College of Pharmacy and Nutrition, University of Saskatchewan, Saskatoon, SK S7N 5A2, Canada; pdc361@mail.usask.ca (A.H.); Gordon.zello@usask.ca (G.A.Z.)

**Keywords:** lentils, chickpeas, beans, peas, exercise, endurance, aerobic, global positioning system

## Abstract

Pulses (i.e., lentils, chickpeas, beans, peas) are low-glycemic index, high-fiber foods that are beneficial for improving blood lipids. Young soccer players typically have low dietary fiber intake, perhaps because of concerns regarding gastro-intestinal problems during exercise performance. Twenty-seven (17 females) soccer players were randomized to receive a pulse-based diet or their regular diet for four weeks in a cross-over study and evaluated for changes in blood lipids and athletic performance, with 19 (22 ± 6y; 12 females) completing the study (eight participants withdrew because of lack of time). Women increased high density lipoproteins (+0.5 ± 0.7 vs. −0.6 ± 0.3 mmol/L; *p* < 0.01) and reduced total cholesterol to high density lipoprotein ratio (−2.4 ± 2.9 vs. +2.6 ± 2.2; *p* < 0.01) on the pulse-based vs. regular diet, respectively, while there were no differences between diet phases in men. Athletic performance assessed by distance covered during games by a global positioning system was not significantly different during the pulse-based vs. regular diet (9180 ± 1618 vs. 8987 ± 1808 m per game; *p* = 0.35). It is concluded that a pulse-based diet can improve blood lipid profile without affecting athletic performance in soccer players.

## 1. Introduction

Low-glycemic index carbohydrates may be superior to high-glycemic index carbohydrates for performance if consumed before endurance exercise; however, the evidence supporting this contention from meta-analyses of studies is deemed to be “weak” [1]. Compared to high-glycemic index carbohydrates, the consumption of low-glycemic index carbohydrates approximately two hours before exercise results in lower insulin release measured immediately prior to exercise [2,3]. Insulin inhibits fat oxidation [4]; therefore, low-glycemic index carbohydrates may be superior to high-glycemic index carbohydrates for stimulating fat oxidation [5], reducing carbohydrate oxidation [2] and preserving muscle glycogen [3], a fuel source thought to be limiting for sports involving repeated bouts of intermittent exercise, including soccer [6]. The effect of low-versus high-glycemic index carbohydrate consumption before intermittent exercise, such as soccer performance, is mixed, with some studies showing no difference between low- and high-glycemic index foods [2,3,5] and others showing that the consumption of low-glycemic index foods (i.e., low-glycemic index sports bars) are beneficial for improving performance (i.e., timed agility running and jumping height when heading) late in a soccer game, compared to high-glycemic index foods [7].

Pulses (which include non-oil seed legumes such as lentils, chickpeas, beans, and split peas) have a very low-glycemic index and are very high in dietary fiber. The high dietary fiber makes pulses a healthy food choice, because when pulses are consumed on a chronic basis (i.e., 8–16 weeks), they reduce harmful blood lipids [8,9], a major risk factor for cardiovascular disease. A number of studies evaluating diet in well-trained soccer players (males and females ranging in age from 16–22 years across studies) indicate that soccer players have relatively low intake of dietary fiber (i.e., about 55–67% of the recommended/reference intake) [10,11] and this may put them at risk of development of cardiovascular disease later in life [12]. It has been proposed that soccer players (mean age of 20 years) may avoid foods high in dietary fiber for fear it might cause gastrointestinal discomfort during training or games [13]. We have conducted a number of lab-based soccer simulations (i.e., on a treadmill), demonstrating that consumption of low-glycemic index pulses (i.e., lentils) 2–3 h before exercise improves metabolic response during exercise (i.e., increases fat oxidation, reduces carbohydrate oxidation), without any negative effects on performance compared to when high-glycemic index foods are consumed [2,3,5]. We therefore propose that athletes, such as soccer players, can derive health benefits from a low-glycemic index pulse-based diet without negative effects on performance.

The purpose of our study was to determine the effects of a month-long pulse-based diet on blood lipids and athletic performance in club-level and university soccer players. We hypothesized that a pulse-based diet would reduce harmful blood lipids, but have no negative impacts on athletic performance.

## 2. Materials and Methods

This study received ethical approval from the University of Saskatchewan Biomedical Research Ethics Board (Bio #14–82), and all participants signed an informed consent form prior to participating in the study.

Twenty-seven (17 females) club-level and university soccer players (mean ± standard deviation [SD] age 22.06 ± 6.12 y) were randomized (by a computer-generated random numbers program) to receive either a pulse-based diet or to continue on their regular diet for four weeks, after which they were to have a two-week washout period, and cross over to the other diet for four weeks.

The pulse-based diet involved the consumption of two servings per day of pulses (i.e., lentils, chickpeas, beans, or split peas) in soups, salads or meals in an amount equal to 25% of participants’ daily caloric intake, determined by 3-day food diaries at baseline. This was equal to about 156 g dry weight (2.4 g/kg body weight), or about 260 g wet weight cooked (4 g/kg body weight), pulses per day. We have previously determined that this amount of pulses is effective for reducing blood lipids in clinical populations [8,9]. All meals were prepared in our university kitchen laboratory and delivered to participants. Two hours before soccer games, players were given lentil-based sports nutrition bars (Genki Foods Inc., Winnepegosis MB, Canada) (i.e., during the pulse-based diet phase) or high glycemic index sports nutrition bars (Clif Bar Inc., crunchy peanut butter flavor, Berelely, CA, USA) (i.e., during the regular diet phase). The high-glycemic index bar was chosen to match the lentil bar as closely as possible for macronutrients. The number of bars given provided 1 g/kg available carbohydrate (i.e., carbohydrate minus fiber). This was equivalent to approximately 3 sports nutrition bars per participant. The glycemic index of the lentil bar is 45 [14] and the glycemic index of the Clif bar is 101 [15]. For a 70 kg individual, the lentil-based bars provided 505 kcal, 12.7 g fat, 85 g total carbohydrate, 70 g available carbohydrate, 15 g fiber, and 26 g protein; whereas the high-glycemic index bars provided 507 kcal, 13.3 g fat, 77 g total carbohydrate, 70 g available carbohydrate, 7 g fiber, and 21 g protein.

The day before and the day after each diet phase, the following assessments were made: serum lipids and insulin were determined from a 20 mL blood sample taken from an antecubital vein the morning after a 12 h fast. The blood was allowed to clot for 30 min and spun in a centrifuge for 10 min at 3000 revolutions per minute, with serum stored at minus 80 °C. An LX20 Beckman Coulter analyzer (Beckman Coulter Canada Inc., Mississauga, ON, USA) was used to analyze the following by enzymatic kits: total cholesterol, low density lipoprotein cholesterol + very low density lipoprotein cholesterol, and high density lipoprotein cholesterol (EnzyChrom™ HDL and LDL/VLDL Assay Kit, EHDL-100, BioAssay Systems, Hayward, CA, USA). Plasma insulin (Alpco Diagnostics, Salem, NH, USA) was measured using a high-sensitivity enzyme-linked immunosorbent assay (ELISA). Samples from all of the time points for each participant were analyzed in the same assay to eliminate between-assay variability. Within assay coefficients of variation ranged from 1.7% to 7.6%. Body composition was determined by dual energy X-ray absorptiometry (Hologic Discovery, Bedford, MD, USA). This included whole-body fat mass, lean tissue mass (bone-free), and percentage body fat. The coefficients of variation for fat mass and lean tissue mass from our lab are 3.0% and 0.5%, respectively.

Match play demands were determined by a global position tracking system (Catapult Optimeye X4 system, Catapult Innovations, Melbourne, Australia), which sampled at 10 Hz [16]. This involves the player wearing a small portable lightweight device that fits into a harness between the shoulder blades (i.e., between the upper scapulae, between the junction of the third and fourth thoracic vertebrae) [17] under their shirt during soccer play. Athletic performance was assessed during match play during the fourth week of each diet phase. Assessment of matches was during the afternoon (Saturdays), which assured that the effects of circadian rhythms on intermittent sport performance between matches was minimal [18]. Variables assessed included total distance covered during the match, maximal velocity, and percent of time resting/walking/jogging (<14.4 km/h), running (14.4–19.7 km/h), and sprinting (≥19.8 km/h) [17]. These were based relative to the fastest players on the teams who averaged (for males and females) 24.8 km/h for fastest velocity. Velocity for sprinting was set at approximately 80% of this velocity and velocity for running at 60–79% of this velocity [17]. These velocity zones are approximately equal to those of others who have assessed time-motion analyses of soccer performance [19]. In addition, we assessed player load during matches, which represents the sum of accelerations and decelerations during a match and gives an indication of the work done [17]. Player workload values are presented as the individual X, Y and Z anatomical plane components and arbitrary units [17]. Exercise for the 24-h before the game-day data collection was consistent throughout the study, consisting of the same low-intensity practice on the Friday before the Saturday games, and included work on technique, strategies, and set pieces.

Twenty-four hour food recalls were collected over three consecutive days (Thursday, Friday, Saturday) during the last week of each diet phase (where games were played on Saturday afternoons) and were analyzed by ESHA Food Processor SQL Software (version 7.02, ESHA Research, Salem, OR, USA).

Our sample size was based on expected changes in harmful lipids, relative to changes in healthy lipids (i.e., total cholesterol to high density lipid ratio). In our most recent clinical study [9], we assessed the same pulse-based diet as used in the current study and found that the pulse-based diet reduced total cholesterol to high density lipid ratio by 0.4 (SD 0.4), compared to an increase of 0.1 (SD 0.4) on a regular diet. With an alpha of 0.05, a power of 80% and a correlation between measurements of 0.7, our sample size estimate was six for the current cross-over study. We therefore considered a sample size of 10 (males and females combined) who were assessed for blood lipids (see results), to be adequate for the current study. We therefore considered a sample size of 10 (males and females combined) to be adequate for the current study.

Blood lipid and body composition data were analyzed by a sex (male vs. female) × diet (pulse vs. control) × time (baseline vs. 4 weeks) ANOVA, with repeated measures on the last two factors. Performance and dietary variables were assessed by sex (male vs. female) × diet (pulse vs. control) ANOVA, with repeated measures on the diet factor. Tukey post-hoc tests were conducted when there were significant interactions. Significance was accepted at *p* ≤ 0.05. All results are presented as means (SD).

## 3. Results

Nineteen (12 females) participants completed the study. Three females and one male dropped out during the control diet phase, and two females and two males dropped out during the pulse diet phases, due to lack of time. There were no reports of adverse events. Out of the 19 participants who completed the study, 10 participants (5 females) agreed to have blood drawn for lipid analyses. Valid performance data were collected from 14 participants (7 females) in each diet phase. Body composition data were collected on all 19 participants during each diet phase.

### 3.1. Lipids

Blood lipids and insulin (mean and SD) are presented in Table 1 below. There was a sex × diet × time interaction for high density lipoproteins (*p* = 0.004). Post-hoc testing indicated that women had significantly higher HDL on the pulse-based diet, compared to the control diet at four weeks (*p* = 0.023; Table 1). There was a sex × diet × time interaction for total cholesterol to HDL ratio (*p* = 0.001). Post-hoc testing indicated that women had a significant decrease while on the pulse-based diet (*p* = 0.010) and a significant increase while on the control diet (*p* = 0.006; Table 1). There were no main effects or interactions for total cholesterol or low density lipoprotein + very low density lipoprotein concentrations (Table 1).

### 3.2. Body Composition

There were no diet × time or diet × sex × time interactions for body composition (Table 2). There were sex main effects, with men having lower percent fat (*p* < 0.001) and fat mass (*p* = 0.05), and higher lean tissue mass (*p* < 0.001) and body mass (*p* = 0.04) compared to women. There were time main effects, with percentage fat decreasing (*p* = 0.016), fat mass decreasing (*p* = 0.009), and lean tissue mass increasing (*p* = 0.04), which could be expected, since participants were assessed while their soccer seasons progressed.

### 3.3. Athletic Performance

There were no differences in performance between diet phases or between sexes. During the pulse-diet phase players covered a mean (SD) distance of 9180 (1618), compared to 8987 (1808) m during the control-diet phase. During the pulse-diet phase, maximal velocity was 25 (3), compared to 27 (3) km/h during the control-diet phase. Time spent resting/walking/jogging was 91 (6)% during the pulse-diet phase, vs. 92 (5)% during the control-diet phase. Time spent running was 7 (3)% during the pulse-diet phase, versus 6 (3)% during the control-diet phase. Time spent sprinting was 2 (2)% during the pulse-diet phase vs. 2 (1)% during the control diet phase. Player loads (arbitrary units) during matches were 875 (203) during the pulse-diet, versus 809 (268) during the control-diet phase.

### 3.4. Dietary Intake

There was no difference between diet phases or between sexes for any of the dietary components, except total fiber intake, which was higher on the pulse-based diet vs. the control diet (41 ± 17 vs. 26 ± 16 g/d; *p* < 0.001). Means (SD) for dietary components (sexes combined) during the pulse-based versus control diet were 2012 (708) vs. 2014 (707) kcal/d, 96 (34) vs. 100 (30) g/d protein, 71 (37) vs. 69 (35) g/d fat and 253 (105) vs. 254 (93) g/d carbohydrate. There were no differences between the three days of diet data collection (i.e., Thursday, Friday, Saturday) for any dietary variables (data not shown).

## 4. Discussion

The main findings were that a low-glycemic index pulse-based diet significantly improved lipid profile (i.e., increased high density lipoproteins and decreased total cholesterol to high density lipoprotein ratio) compared to a regular diet in female, but not male soccer players. There was no effect on insulin. We can only speculate on why females improved their lipid profile, while men did not. Females generally have a more favorable lipid metabolism compared to males; this may be related to higher estrogen concentrations [20]. Perhaps a higher concentration of estrogen allows one to be more responsive to any intervention that would improve lipid profile. Despite the high fiber content of the pulse-based diet, there were no negative effects on athletic performance. This is important, as young soccer players typically have a low intake of dietary fiber, and it is proposed that some might avoid fiber, for fear of gastrointestinal discomfort during training or games [10,11,12,13]. The findings of the current study do not support this assumption and indicate that low-glycemic index high-fiber foods such as pulses can be a healthy component of an athlete’s diet without negatively affecting performance. This is supported by our previous lab-based studies, where the consumption of pulses before soccer-simulated treadmill sessions did not negatively affect performance compared to high-glycemic index foods [2,3,5]. The strength of the current study is that these findings have been extended to the chronic consumption of pulses and actual on-field soccer performance.

Blood lipids were favorably affected while on the pulse-based versus regular diet; this is supported by our previous work in clinical populations [8,9]. An identical pulse-based diet, as used in the current study, was effective for increasing the high density lipoprotein concentration and reducing total cholesterol to high density lipoprotein ratio in women with polycystic ovary syndrome over 16 weeks [9], and reducing total cholesterol and low density lipoprotein concentration in older men and women (mean age 59.7 years) over eight weeks [8]. It is unclear why the pulse-based diet only had a positive effect on females, but not males, in the current study. This may be due to the low participant numbers and high variation in the current study, leading to the lack of adequate statistical power.

Insulin concentrations were unchanged during our intervention. This may be due to the young, healthy status of our participants, who would be less responsive to a low glycemic index diet compared to clinical populations. For example, in previous work with women with endocrine abnormalities (i.e., polycystic ovary syndrome), a longer (16 weeks) low-glycemic index pulse-based diet (using identical meals as the current study) reduced insulin concentrations during an oral glucose tolerance test [9]. Our previous studies in soccer players found that insulin levels were lower during postprandial measurements, after acute intake of pulse-based foods (i.e., lentils), compared to foods with a higher glycemic index [2,3,7]. In the current study, we did not find an effect on fasting insulin levels with chronic intake of a pulse-based diet.

Reduced total cholesterol to high density lipoprotein concentration during the pulse-based diet may be due to the high fiber content of the diet. Fiber binds to bile acids, decreasing their reabsorption, leading to increased hepatic bile acid production, which reduces the hepatic cholesterol pool, necessitating an increased cholesterol uptake from the blood into the liver [21]. Additionally, the fermentation of soluble fiber in the colon produces short chain fatty acids which stimulate a decrease in hepatic cholesterol synthesis [21]. There was no effect of the pulse-based diet on body composition in the current study. This is in contrast to our previous study in older adults on an identical pulse-based diet, where those with a higher than normal body fat percentage had a significant decrease in body fat percentage on the pulse-based versus regular diet [8]. Differences between studies could be accounted for by the very different populations assessed (i.e., inactive older adults versus very active young soccer players) and the fact that the intervention in the older adults was twice as long (i.e., eight weeks, compared to four weeks in the current study).

There were no differences in performance measures between low and high-glycemic index diet conditions, as assessed by distance covered during the games, percentage of time spent resting, walking, jogging, running, or sprinting, and player load. The percentage of time spent resting, walking, jogging, running, and sprinting in our participants is very similar to that from time motion analyses of games in other studies [19]. We have previously assessed lab-based simulations of soccer games on a treadmill to simulate these speeds [3,5], during which we assessed heart rate and substrate utilization by measuring metabolic gases (i.e., oxygen consumption and carbon dioxide output). These speeds elicited an average heart rate during games, which was 70% of peak heart rate [5], carbohydrate oxidation rates of 2.8–3.2 g/min and fat oxidation rates of 0.05–0.17 g/min [3,5]. Since treadmill velocities used in these soccer game simulations were similar to what was assessed by the global position tracking system in the current study, we can assume the heart rates and substrate oxidation rates would have been similar in the games assessed in the current study. The relatively high carbohydrate oxidation rates during these soccer games could be related to catecholamine response, which increases throughout games, especially after consumption of high-glycemic index carbohydrate [3]. This may lead to greater glycogen breakdown due to the catecholamine activation of glycogen phosphorylase [3]. This comparison is limited, however, because data obtained from a treadmill simulation cannot perfectly match actions of a soccer match, where repeated bouts of intermittent exercise are interspaced with different actions, such as continuous changes of direction, accelerations and decelerations.

## 5. Conclusions

A low-glycemic index pulse-based diet was beneficial for improving serum lipid profile in female but not male soccer players; however, the low-glycemic index diet did not have a beneficial effect on performance in the current study compared to the high-glycemic index diet.

## Figures and Tables

**Table 1 nutrients-12-01324-t001:** Blood Lipids and Insulin.

	Women				Men			
	Pulse Diet		Control Diet		Pulse Diet		Control Diet	
	Baseline	Four Weeks	Baseline	Four Weeks	Baseline	Four Weeks	Baseline	Four Weeks
LDL + VLDL (mmol/L)	2.0 (0.8)	2.1 (0.3)	2.1 (0.4)	1.6 (0.6)	1.8 (0.7)	1.8 (0.5)	1.8 (0.3)	1.7 (0.2)
TC (mmol/L)	2.0 (0.2)	1.9 (0.2)	1.9 (0.1)	1.9 (0.2)	1.9 (0.4)	1.8 (0.4)	1.7 (0.3)	2.0 (0.4)
HDL (mmol/L)	0.8 (0.6)	1.3 (0.7) *	1.2 (0.2)	0.6 (0.4)	1.2 (0.4)	0.8 (0.3)	1.3 (0.3)	1.0 (0.6)
TC/HDL	4.1 (3.0)	1.7 (0.6) *^,‡^	1.6 (0.4)	4.3 (2.5) ^‡^	1.6 (0.5)	2.4 (1.1)	1.5 (0.4)	2.7 (1.7)
Insulin (µIU/mL)	6.0 (3.8)	7.6 (2.2)	5.4 (2.6)	6.3 (2.5)	7.5 (4.6)	4.0 (4.6)	6.4 (1.9)	4.1 (3.0)

Data are means (SD); *n* = five women and five men; LDL = Low density lipoproteins; VLDL = very low density lipoproteins; TC = Total cholesterol; HDL = high density lipoproteins; * Significant difference for pulse-based versus control diet at four weeks (*p* < 0.05; Tukey post-hoc test on a Sex × Diet × Time interaction; *p* < 0.01); ^‡^ Significant difference from baseline to four weeks within the diet condition (*p* ≤ 0.01; Tukey post-hoc test on a Sex × Diet × Time interaction; *p* < 0.01).

**Table 2 nutrients-12-01324-t002:** Body Composition.

	Women ^‡^				Men			
	Pulse Diet		Control Diet		Pulse Diet		Control Diet	
	Baseline	Four Weeks	Baseline	Four Weeks	Baseline	Four Weeks	Baseline	Four Weeks
LTM (kg) *	45.2 (5.1)	45.8 (5.1)	45.6 (5.3)	46.0 (5.2)	59.3 (7.3)	59.6 (7.0)	59.0 (6.7)	59.7 (8.0)
FM (kg) *	16.2 (4.2)	14.7 (4.1)	15.3 (4.4)	14.8 (4.8)	11.2 (3.8)	10.6 (3.5)	11.4 (4.7)	11.1 (3.9)
% Fat *	24.9 (2.9)	23.2 (4.1)	23.9 (4.1)	23.1 (4.6)	15.1 (2.9)	14.1 (2.8)	15.1 (4.1)	14.9 (2.9)
TBM (kg)	61.5 (8.4)	60.6 (7.6)	60.8 (8.6)	60.9 (8.4)	70.6 (10.8)	70.2 (10.4)	70.4 (10.7)	70.7 (11.7)

Data are means (SD); *n* = 12 women and 7 men; LTM = Lean tissue mass; FM = Fat mass; TBM = Total body mass; * Time main effects (*p* < 0.05); ^‡^ Sex main effect for each variable (*p* ≤ 0.05).

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
