# Peer review of "A Low-Glycemic Index, High-Fiber, Pulse-Based Diet Improves Lipid Profile, but Does Not Affect Performance in Soccer Players"

_nutrients, 2020, doi:10.3390/nu12051324_

Round 1

Reviewer 1 Report

The authors executed a sound research design with relevant results/conclusions. In that regard the authors were successful. However, it is important to note that substrate utilization is most influenced by exercise intensity. However, as indicated within the background, substrate utilization is strongly influenced by hormone status, and therefore exercise intensity. This concept is absent within the current manuscript. With blood samples for each condition, I recommend that if enough volume was collected to include hormone analysis (ELIZA) and exercise intensity (HR). If the portable device include HR exercise intensity can be derived indicating reliance on a particular substrate, that will significantly enhance the manuscript rigor. Furthermore, substrate utilization can be influenced by 24hrs of diet (Siliva et al. 2016). 3dy diet recall is monitored but it is not clear whether it was completed within 24hrs of blood collection/testing. Lastly, was diet monitored after exercise between time points? Nutrient specificity within 24hrs post exercise can inlfluence substrate utilization (Pilegaard et al. '14-'16)

Relevant hormones for consideration would be included within the analysis would be insulin and epinephrine, both of which are influenced by diet and acute/chronic exercise. Including these within the analysis is necessary to better understand substrate utilization during activity and post activity 

The discussion/conclusions are relevant to the results presented. However, the manuscript is limited due to methodological considerations that need to be rectified to meet the Journal rigor/Nutrients readership. 

Author Response

Thank you for the review of our manuscript. We have copied each of the reviewer’s comments below, followed by our responses. All revisions in the manuscript are highlighted in yellow.

The authors executed a sound research design with relevant results/conclusions. In that regard the authors were successful. However, it is important to note that substrate utilization is most influenced by exercise intensity. However, as indicated within the background, substrate utilization is strongly influenced by hormone status, and therefore exercise intensity. This concept is absent within the current manuscript. With blood samples for each condition, I recommend that if enough volume was collected to include hormone analysis (ELIZA) and exercise intensity (HR). If the portable device include HR exercise intensity can be derived indicating reliance on a particular substrate, that will significantly enhance the manuscript rigor.

Response: The portable device does not include HR exercise intensity; however, the device gave an indication of intensity as percentage of time spent during the game resting, walking, jogging, running, and sprinting. The percentage of time spent in each of these domains is presented in the results section. We can relate this to HR intensity because we have previously run studies in our lab on a treadmill with varying speeds to simulate soccer matches with similar percentages of time spent at each of these intensities (Little et al. 2009; 2010). These tests elicited an average HR during simulated soccer games that was about 70% of HR max. This resulted in mostly reliance on carbohydrate oxidation (about 2.8-3.2 g/min) and smaller amounts of fat oxidation (about 0.05-0.17 g/min). We have added this to the discussion section to give a better idea of the intensity elicited during games and the most likely substrates oxidized.

Furthermore, substrate utilization can be influenced by 24hrs of diet (Siliva et al. 2016). 3dy diet recall is monitored but it is not clear whether it was completed within 24hrs of blood collection/testing.

Response: The dietary recall was done on three consecutive days including two weekdays and one weekend day. Games were played on Fridays, Saturdays, and Sundays; therefore, most dietary recall data would have been within 24 hours of games (and therefore testing). This has been added to the methods section where we describe dietary monitoring.

Lastly, was diet monitored after exercise between time points? Nutrient specificity within 24hrs post exercise can inlfluence substrate utilization (Pilegaard et al. '14-'16)

Response: As mentioned above, dietary data would have been collected Thursday, Friday, and Saturday and games played Friday, Saturday or Sunday; therefore, much of the dietary data was collected before and/or after games.

Relevant hormones for consideration would be included within the analysis would be insulin and epinephrine, both of which are influenced by diet and acute/chronic exercise. Including these within the analysis is necessary to better understand substrate utilization during activity and post activity

Response: We have added insulin data to table 1. There was the expected trend with insulin tending to be reduced during low glycemic index diets across all participants; however, differences between conditions was not statistically significant.

Although we did not assess epinephrine in the current study, we have previously assessed epinephrine during lab-based (treadmill) simulations of soccer games (Little et al. 2010). As mentioned above, the velocities used during this treadmill testing was very similar to that experienced during games in the current study. We have added to the discussion section that catecholamines were increased during simulated games in our previous studies, especially after consumption of high glycemic index carbohydrate, and this may lead to activation of glycogen phosphorylase and greater break down of glycogen.

The discussion/conclusions are relevant to the results presented. However, the manuscript is limited due to methodological considerations that need to be rectified to meet the Journal rigor/Nutrients readership.

Response: Thank you for your helpful comments and your time commitment to the peer-review of our manuscript. We have tried to address each comment (above) and revised our manuscript accordingly.

Reviewer 2 Report

Many thanks for the opportunity of reviewing this article. The rationale for your study is well presented, though there are important weaknesses in this paper. First of all, in your abstract section you mentioned that your sample size is 19 young soccer players (12 females), however in your Methods section you mentioned 27 (17 females). Please clarify.  In addition, you have identified a reduced sample size (n = 19). Did you do sample size calculations for your measures of interest prior to running this study?

Abstract: 

Line 13-14: The authors mentioned "fiber be avoided by endurance athletes in preparation for competition because it is fearedthey may lead to gastro-intestinal problems and impaired sport performance". However your study is realized in an intermittent sport such as soccer, do you think that the effects in soccer would be the same that in endurance sports with this type of diet?

Line 17: Different letter style "athletic".

Line 21: Simulated soccer games? In addition, please details of GPS model.

Introduction

The introduction is limited and would benefit from more background. Specifically, I would encourage the authors to highlight the specific effects of high GI vs low GI in sports performance.

Line 28: The authors stated "Low-glycemic index carbohydrates may be superior to high-glycemic index carbohydrates for performance if consumed before exercise". However, according to Wong et al (doi: 10.1093/nutrit/nux003) "weak evidence" is reported in the meta-analysis. Please clarify.

Line 34: Soccer in an endurance sport?. Please see. (doi:10.1080/02640410500482529)

Material and Methods

The Methods section is very limited specially in the match play demands measured by GPS. I would like to encourage the authors to highlight more information about the match play demands.

Line 55: Twenty seven participants? See the Abstract section. Please add two decimals to mean+SD.

Line 65: Please details quantity (sport nutrition bars).

Line 90: Please add harness details.

Lines 91-92: The authors stated "Athletic performance was assessed during
match play during the fourth week of each diet phase" All the matches were played at the same time-of-day to avoid the detrimental effects associated to circadian rhythms? If all the soccer matches were realized at the same time-of-day added reference in intermittent sports (doi: 10.1080/02640414.2016.1258481) 

Lines 92-94: Why you selected this speed zones. "included total distance covered during the match, maximal velocity, and percent of time resting/93 walking/jogging (<14.49 km/h), running (14.4-19.7 km/h), and sprinting (≥19.8 km/h)". In addition, please add to manuscript the acceleration and decceleration zones. 

Results:

Lines 114-116: Please be consistent with decimals.

Lines 139-141: Rewording.

Discussion:

Lines 149-151: Could the authors hypothetized why female significantly improves lipid profile?.

Lines 151-152: Such as previously mentioned soccer is an intermittent sport not an endurance sport. Please clarify.

Conclussion:

The authors stated "A low-glycemic index pulse-based diet was beneficial for improving serum lipid profile in female, but not male soccer players. The high fiber content of the diet did not have any negative effects on athletic performance". My question is; low glycemic index improved sports performance according to the data obtained in your study? Please clarify.

Author Response

Thank you for the review of our manuscript. We have copied each of the reviewer’s comments below, followed by our responses. All revisions in the manuscript are highlighted in yellow.

Many thanks for the opportunity of reviewing this article. The rationale for your study is well presented, though there are important weaknesses in this paper.

First of all, in your abstract section you mentioned that your sample size is 19 young soccer players (12 females), however in your Methods section you mentioned 27 (17 females). Please clarify.  In addition, you have identified a reduced sample size (n = 19). Did you do sample size calculations for your measures of interest prior to running this study?

Response: We have clarified this in the abstract – 27 players were initially randomized, and 19 players completed the cross-over study.

The sample size calculation was based on expected change in ratio of harmful to healthy lipids (i.e. Total cholesterol to HDL ratio). In our most recent clinical study (Kazemi et al. 2018) we assessed the same pulse-based diet as used in the current study and found that the pulse-based diet reduced Total cholesterol to HDL ratio by 0.4 (SD 0.4) compared to an increase of 0.1 (SD 0.4) on a regular diet. With an alpha of 0.05, a power of 80% and a correlation between measurements of 0.7, our sample size estimate is 6 for a cross-over study. We were able to assess 10 of the soccer players for changes in blood lipids; therefore, we feel our sample size was adequate. We have added the sample size calculation before the statistics section in the manuscript.

Abstract:

Line 13-14: The authors mentioned "fiber be avoided by endurance athletes in preparation for competition because it is feared they may lead to gastro-intestinal problems and impaired sport performance". However your study is realized in an intermittent sport such as soccer, do you think that the effects in soccer would be the same that in endurance sports with this type of diet?

Response: In response to the reviewer’s comments on continuous endurance exercise versus the intermittent exercise performed by soccer players, we did a literature search on soccer players and dietary fiber intake. We found that young soccer players typically have a very low dietary fiber intake and studies suggest this might be due to fear of gastro-intestinal discomfort during exercise. We have modified the wording in our abstract and elsewhere in the manuscript to reflect these studies of soccer players.

Line 17: Different letter style "athletic".

Response: We have corrected the font style here to match the font style of the manuscript.

Line 21: Simulated soccer games? In addition, please details of GPS model.

Response: The games we evaluated were not simulated. They were actual games during a collegiate season. We have provided the details of the GPS model, as requested.

Introduction

The introduction is limited and would benefit from more background. Specifically, I would encourage the authors to highlight the specific effects of high GI vs low GI in sports performance.

Response: We have added greater detail regarding sport performance (specifically soccer performance), in response to high versus low GI foods.

Line 28: The authors stated "Low-glycemic index carbohydrates may be superior to high-glycemic index carbohydrates for performance if consumed before exercise". However, according to Wong et al (doi: 10.1093/nutrit/nux003) "weak evidence" is reported in the meta-analysis. Please clarify.

Response: In the introduction we have added that the evidence from this meta-analyses was deemed to be weak, as pointed out by the reviewer.

Line 34: Soccer in an endurance sport?. Please see. (doi:10.1080/02640410500482529)

Response: We have changed the terminology here when referring to soccer from “endurance sport” to “sports involving repeated bouts of intermittent exercise” when referring to the soccer studies we have referenced in this section.

Material and Methods

The Methods section is very limited specially in the match play demands measured by GPS. I would like to encourage the authors to highlight more information about the match play demands.

Response: We have added “player load” to our GPS results, which measures the sum of all accelerations and decelerations during a match and gives an indication of work done during the match. In addition to player load, we have presented total distance covered, maximal velocity, and percentage of time spent walking, jogging, running, and sprinting, which gives a good indication of match play demands. This has been added to the methods section and the results section (i.e. section 3.3).

Line 55: Twenty seven participants? See the Abstract section. Please add two decimals to mean+SD.

Response: We have clarified the participant numbers in the abstract. We have added two decimals to the mean and SD as requested by the reviewer.

Line 65: Please details quantity (sport nutrition bars).

Response: The approximate number of sports nutrition bars to achieve 1g/kg available carbohydrate was three. This has been added to the methods section of the manuscript.

Line 90: Please add harness details.

Response: We have added more detail regarding the harness.

Lines 91-92: The authors stated "Athletic performance was assessed during match play during the fourth week of each diet phase" All the matches were played at the same time-of-day to avoid the detrimental effects associated to circadian rhythms? If all the soccer matches were realized at the same time-of-day added reference in intermittent sports (doi: 10.1080/02640414.2016.1258481)

Response: The matches were all played either during Friday evening, or Saturday/Sunday afternoon. Since differences in circadian rhythms for intermittent sports have been identified for morning versus afternoon sessions (according to the reference provided by the reviewer) we believe circadian rhythms would have minimal effect on our results. We have added a statement in our methods on this in the manuscript, and have cited the reference provided by the reviewer.

Lines 92-94: Why you selected this speed zones. "included total distance covered during the match, maximal velocity, and percent of time resting/93 walking/jogging (<14.49 km/h), running (14.4-19.7 km/h), and sprinting (≥19.8 km/h)". In addition, please add to manuscript the acceleration and decceleration zones.

Response: The speed zones were selected relative to the fastest players on the men’s and women’s soccer teams, who ran at a top speed of 24.8 km/h. The sprinting speed was set at 80% of this velocity and the running speed at approximately 60-79% of this velocity. This is very similar to the sprinting and running speed selected by others who have performed time-motion analyses of soccer players (i.e. Ali and Farrally, 1991). We have added these details to the manuscript. We did not analyze specific acceleration and deceleration zones per se. We have added to the manuscript the variable of player load from the GPS which gives an indication of the sum of accelerations and decelerations during a match.

Results:

Lines 114-116: Please be consistent with decimals.

Response: We have corrected the number of decimal places presented here to be consistent

Lines 139-141: Rewording.

Response: We have reworded this section as suggested by the reviewer.

Discussion:

Lines 149-151: Could the authors hypothetized why female significantly improves lipid profile?.

Response: Females generally have a more favorable lipid metabolism compared to males; this is possibly related to higher estrogen levels (Palmisano et al. 2018). Perhaps a higher level of estrogen allows one to be more responsive to any intervention that would improve lipid profile. We have added this to our discussion section.

Lines 151-152: Such as previously mentioned soccer is an intermittent sport not an endurance sport. Please clarify.

Response: We have modified this section to reflect that young soccer players typically have low fiber intakes, possibly because they avoid fiber out of fear of gastrointestinal problems.

Conclussion:

The authors stated "A low-glycemic index pulse-based diet was beneficial for improving serum lipid profile in female, but not male soccer players. The high fiber content of the diet did not have any negative effects on athletic performance". My question is; low glycemic index improved sports performance according to the data obtained in your study? Please clarify.

Response: The low glycemic index diet did not improve sports performance according to our data. This has been added to the conclusion section.

Round 2

Reviewer 1 Report

Thank you to the authors for a fine attempt at a revision and for addressing concerns previously cited. However, after careful review and in consideration of the inclusion of  insulin data the conclusions need to be re-considered. Data table 1 is most interesting as the means for each group/metric do not tell the same story as the author's conclusions. Further, in consideration of the background literature related to insulin inhibition of lipid oxidation, the finds do not support the conclusions. "Insulin inhibits fat oxidation [4]; therefore, low-glycemic index carbohydrates may be superior to high-glycemic index carbohydrates for stimulating fat oxidation [5], reducing carbohydrate oxidation [2] and preserving muscle glycogen [3], a fuel source thought to be limiting for endurance exercise, including soccer [6]."

Abstract:

Does Nutrients allow name brands included within the abstract? In some cases this is considered a breach of etiquette to market such products reducing the efficacy of the study objectivity.

Introduction:

Ln 31-33- please include timeline relevant to decreased insulin prior to exercise; e.g. 24hrs, 4hrs, 1hr.

Ln33-36- please provide timeline for context. Current sport nutrition guidelines recommend low glycemic index foods >4hrs prior to exercise. The authors assertions within this paragraph are correct in some circumstances. However, more specificity I believe will calcify any discrepancy.

Ln46-48- what age group and training status have low dietary fiber intake? Please be specific.

Ln48- “young soccer players” please be specific.

Ln52- “before exercise” how long before?

Much of the background discusses the literature from an acute intervention perspective while the purpose of the study is using a chronic dietary approach. I suggest including more background literature using longitudinal data (population specific or not) to strengthen the justification of study and relevant literature.

Ln118-120- what percentage diet recall were within 24hrs of testing? For those that exceeded 24hrs, how did their diet differentiate, if at all? What type of dietary trends might have disrupted normal eating  prior to testing e.g.: Team meals, late night snacks. How can the authors ensure no radicle shifts in dietary intake occurred. Also, substrate utilization can be affected by exercise within 24hrs. Please include pre test exercise guidelines as standardized provision if this was standardized.https://www.researchgate.net/publication/13625016_Substrate_use_during_and_following_moderate-_and_low-intensity_exercise_Implications_for_weight_control

Results:

Ln144- please include “Blood lipids and insulin mean+SD values are presented in table 1 below.”

Table 1: was insulin included within the stat analysis? Female Pulse diet Insulin had a non-significant 21% increase post intervention? This is contradictory to the background literature presented above?

How did female pulse diet LDL+VLDL go up by 0.1, HDL go up by 0.5 and total cholesterol go down by 0.1? Same question with Female Control Diet metrics.

Ln174- please include no differences “in performance” . . .

Discussion:

Ln194-197- Please include findings of insulin where there was an increase. According to the intro/background literature this would suppress lipid utilization.

Ln19-200- Recent literature has shown that estrogen concentrations (despite having an influence on lipid transport proteins) have been not been shown to influence lipid oxidation. https://journals.physiology.org/doi/full/10.1152/japplphysiol.00774.2019

The conclusions need to be strengthened by increasing the discussion regarding sex responses and diet type. Male and females responded drastically different. For instance, male HDL went down in both groups. For the volume of exercise this is a finding alternative to much of the literature. Further, the insulin finds are entirely lost within the discussion. These findings with respect to exercise metabolism is very interesting regarding pre-exercise nutrient intake. Expanding on these points will significantly improve the strength of the finds and therefore contribution to the literature. 

Author Response

Thanks again for reviewing our manuscript. We have responded to each reviewer comment below with reference to line numbers in the revised manuscript. Revisions to the manuscript have been highlighted.

Thank you to the authors for a fine attempt at a revision and for addressing concerns previously cited. However, after careful review and in consideration of the inclusion of insulin data the conclusions need to be re-considered. Data table 1 is most interesting as the means for each group/metric do not tell the same story as the author's conclusions.

Further, in consideration of the background literature related to insulin inhibition of lipid oxidation, the finds do not support the conclusions. "Insulin inhibits fat oxidation [4]; therefore, low-glycemic index carbohydrates may be superior to high-glycemic index carbohydrates for stimulating fat oxidation [5], reducing carbohydrate oxidation [2] and preserving muscle glycogen [3], a fuel source thought to be limiting for endurance exercise, including soccer [6]."

Response: We have added a section in the discussion about the insulin results (line 228). We have changed the conclusion (as suggested by the other reviewer) to more succinctly summarize the results of our study (line 271).

Abstract:

Does Nutrients allow name brands included within the abstract? In some cases this is considered a breach of etiquette to market such products reducing the efficacy of the study objectivity.

Response: The name of the GPS system was added based on a previous comment from the other reviewer. We have removed it from the abstract based on your comment.

Introduction:

Ln 31-33- please include timeline relevant to decreased insulin prior to exercise; e.g. 24hrs, 4hrs, 1hr.

Response: The studies that are cited here measured insulin immediately before exercise. This wording has been added (line 33).

Ln33-36- please provide timeline for context. Current sport nutrition guidelines recommend low glycemic index foods >4hrs prior to exercise. The authors assertions within this paragraph are correct in some circumstances. However, more specificity I believe will calcify any discrepancy.

Response: The studies that are cited here provided low glycemic index meals two hours before exercise. We have added this to this section (line 32).

Ln46-48- what age group and training status have low dietary fiber intake? Please be specific.

Response: The ages across studies ranged from 16-22y. The studies were on elite-level, professional, or collegiate soccer players; therefore, they would have all been well-trained. This information has been added to these lines (line 47).

Ln48- “young soccer players” please be specific.

Response: We have added the age range for the cited studies (16-22y) (line 47)

Ln52- “before exercise” how long before?

Response: In these studies, food consumption was 2-3 hours before exercise. This has been added to the manuscript (line 53).

Much of the background discusses the literature from an acute intervention perspective while the purpose of the study is using a chronic dietary approach. I suggest including more background literature using longitudinal data (population specific or not) to strengthen the justification of study and relevant literature.

Response: We have added the following sentence to include studies that assessed the beneficial health effects of consuming low glycemic index pulses on a chronic basis: “The high dietary fiber makes pulses a healthy food choice because when pulses are consumed on a chronic basis (i.e. 8-16 weeks), they reduce harmful blood lipids [8,9], a major risk factor for cardiovascular disease.” (line 45)

Ln118-120- what percentage diet recall were within 24hrs of testing? For those that exceeded 24hrs, how did their diet differentiate, if at all? What type of dietary trends might have disrupted normal eating prior to testing e.g.: Team meals, late night snacks. How can the authors ensure no radicle shifts in dietary intake occurred. Also, substrate utilization can be affected by exercise within 24hrs. Please include pre test exercise guidelines as standardized provision if this was standardized.https://www.researchgate.net/publication/13625016_Substrate_use_during_and_following_moderate-_and_low-intensity_exercise_Implications_for_weight_control

Response: Our research assistant who collected dietary data has confirmed to us that all diet recalls were done on Thursday, Friday, and Saturday of the last week of each diet phase. For each of these dietary recall periods, games were played on Saturday; therefore, two-thirds of the dietary recall were within 24 hours of the testing (i.e. Friday and Saturday would have been within 24 hours). Diets did not differ across the three days of recall (Thursday, Friday, Saturday). The teams travelled together and their dietary intakes were quite consistent throughout the season, as indicated by the lack of difference between the two 3-day recalls that were done during the two different phases of the study. The pre-game exercise routine was consistent across the participants with the same low-intensity practice done the day before games, where players worked on techniques, strategies, and set pieces. These details have been added to the revised manuscript (lines 120-124).

Results:

Ln144- please include “Blood lipids and insulin mean+SD values are presented in table 1 below.”

Response: This wording has been added to the manuscript (line 153).

Table 1: was insulin included within the stat analysis? Female Pulse diet Insulin had a non-significant 21% increase post intervention? This is contradictory to the background literature presented above?

Response: This increase was not statistically significant. It should be pointed out that females tended to also increase during the regular diet phase and when male and female data were combined they had a numerically greater reduction in insulin concentrations during the low-glycemic index diet compared to the regular diet phase (-1.0 vs. -0.69 µIU/mL) but none of these differences were statistically significant. The relatively large increase for insulin in the females was influenced by one outlier (who had a very low baseline measurement in the pulse phase and an “average” value post-intervention). When we excluded this participant from the statistical analyses, it did not make a difference; therefore, for completeness we have included this individual’s data in the table. It should also be noted that these insulin values are fasted values after a four-week intervention in relatively healthy individuals. In our previous soccer-based studies, we found lower post-prandial insulin levels after pulse-based versus higher-glycemic index diets. This could explain the differences between studies. A section has been added to the discussion on these results and comparisons to other studies (line 228).

How did female pulse diet LDL+VLDL go up by 0.1, HDL go up by 0.5 and total cholesterol go down by 0.1? Same question with Female Control Diet metrics.

Response: We checked the values from the spreadsheets from our assays and confirm that the pre- and post-measurements for each diet phase we reported are correct. Each assay (i.e. HDL, LDL+VLDL, and cholesterol) are different assays with a certain amount of variation within each assay. This may explain the differences that might appear in a relatively small sample of women (n=5).

Ln174- please include no differences “in performance” . . .

Response: We have changed the wording as requested (line 184).

Discussion:

Ln194-197- Please include findings of insulin where there was an increase. According to the intro/background literature this would suppress lipid utilization.

Response: We have added a statement on the insulin response here (i.e. that there was no effect of the low-glycemic index pulse-based diet on insulin). Note that there were no significant changes in insulin (i.e. insulin did not increase). (line 205)

Ln19-200- Recent literature has shown that estrogen concentrations (despite having an influence on lipid transport proteins) have been not been shown to influence lipid oxidation. https://journals.physiology.org/doi/full/10.1152/japplphysiol.00774.2019

Response: We added this statement about the possibility of estrogen affecting lipid responses in response to the other reviewer’s comments on the first draft of our manuscript. They asked us to speculate on why females responded favorably to the intervention compared to males. The reference provided above (https://journals.physiology.org/doi/full/10.1152/japplphysiol.00774.2019) is for a study that assessed the effects of different phases of the menstrual cycle on fat oxidation during exercise. I am not sure that this reference would fit into this discussion on blood lipids.

The conclusions need to be strengthened by increasing the discussion regarding sex responses and diet type. Male and females responded drastically different. For instance, male HDL went down in both groups. For the volume of exercise this is a finding alternative to much of the literature. Further, the insulin finds are entirely lost within the discussion. These findings with respect to exercise metabolism is very interesting regarding pre-exercise nutrient intake. Expanding on these points will significantly improve the strength of the finds and therefore contribution to the literature.

Response: We have mentioned in the conclusions that females responded to the intervention with an improvement in lipid profile compared to males (line 271). The male HDL levels did not go down in both conditions, as these changes were not statistically different across time points.

We have added a paragraph in the discussion on the insulin results (line 228). We explain that fasting insulin concentrations did not change with chronic intake of a low-glycemic index diet probably because our participants were young and healthy. This is in contrast to our previous work with clinical populations (i.e. women with polycystic ovary syndrome) who had a decrease in insulin concentrations (during an oral glucose tolerance test) when on the same diet. Also, our previous soccer studies showed that insulin levels in the postprandial state were lower after consumption of low-glycemic index pulse-based foods (i.e. lentils) compared to higher-glycemic index foods. The current manuscript did not assess post-prandial insulin responses to acute intake of low-glycemic index foods, but instead assessed fasting insulin concentrations during a chronic low-glycemic index pulse-based diet.

Reviewer 2 Report

Many thanks for the opportunity or review this manuscript. The suggested changes have significantly improved the quality of the manuscript. However, a few questions need to be elucidated.

Abstract: 

Line 18: Could the authors explain briefly the reasons for the mortality in the participants during the study?

Line 24: Add p-value.

Introduction

Line 47: The authors stated in the manuscript  "A number of studies evaluating diet in young soccer players indicate that soccer players have relatively low intake of dietary fiber". I would like to know if the authors refers to Recommended Nutrient Intake (RNI)....

Line 48-50 The authors stated "It has been proposed that young soccer players may avoid foods high in dietary fiber for fear it might cause gastrointestinal discomfort 49 during training or games [13]". However, the reference that the authors mentioned is the manuscript is a study realized in University students. Do you think that is possible to compare both age groups? Please explain.

 Methods:

Line 66: Twenty-seven participants finished the study? Please see Abstract section and modify.

Line 73. Please provide data in grams/kg of body weight.

Line 76: How many bars? Please provide quantity data.

Line 99: Change "athletic performance" by "match-play demands".

Line 108: Please add reference.

Line 110: Please add reference.

Discusssion:

Lines 241-243: The authors stated "treadmill velocities used in these soccer game simulations were similar to what was assessed by the global position tracking system in the current study, we can assume the heart rates and substrate oxidation rates would have been similar in the games assessed in the current study". It is possible to compare data obtained during the treadmill test with a soccer match where repeated bouts of intermittent exercise intersped with different actions such as continuous changes-of-direction, accelerations and decelerations are involved? Please explain carefully.

 Conclusions:

The conclusions should be "A low-glycemic index pulse-based diet was beneficial for improving serum lipid profile in female but not male soccer players; however, the low-glycemic index diet did not have a beneficial effect on performance in the current study compared to the high-glycemic index diet".

Author Response

Thanks again for reviewing our manuscript. We have responded to each reviewer comment below with reference to line numbers in the revised manuscript. Revisions to the manuscript have been highlighted.

Many thanks for the opportunity or review this manuscript. The suggested changes have significantly improved the quality of the manuscript. However, a few questions need to be elucidated.

Abstract:

Line 18: Could the authors explain briefly the reasons for the mortality in the participants during the study?

Response: We have added to the abstract that 8 participants withdrew from the study due to lack of time. (line 18)

Line 24: Add p-value.

Response: The p-value has been added (line 24)

Introduction

Line 47: The authors stated in the manuscript "A number of studies evaluating diet in young soccer players indicate that soccer players have relatively low intake of dietary fiber". I would like to know if the authors refers to Recommended Nutrient Intake (RNI)....

Response: We have added reference to the RNI for dietary fiber here from these cited studies. Soccer players in these studies were consuming 55-67% of the RNI for fiber (line 48).

Line 48-50 The authors stated "It has been proposed that young soccer players may avoid foods high in dietary fiber for fear it might cause gastrointestinal discomfort 49 during training or games [13]". However, the reference that the authors mentioned is the manuscript is a study realized in University students. Do you think that is possible to compare both age groups? Please explain.

Response: The soccer players in the referenced study (i.e. reference 13) were university students with mean age 20y, similar to our study. We would consider the age of 20y to be “young”. To avoid confusion, we have deleted the word “young” from this section and have instead indicated the ages of the players in the studies we cite. (lines 47 and 50)

Methods:

Line 66: Twenty-seven participants finished the study? Please see Abstract section and modify.

Response: We have revised the wording in our methods here so that it is not implied that 27 participants finished the study. We report here the original number of participants randomized and then in the results indicate that eight of these participants dropped out of the study due to lack of time. (lines 70 and 143)

Line 73. Please provide data in grams/kg of body weight.

Response: These data have been added. (line 75)

Line 76: How many bars? Please provide quantity data.

Response: This information has been added. On average participants had to eat 3 bars to consume 1g/kg available carbohydrate. (line 83)

Line 99: Change "athletic performance" by "match-play demands".

Response: The wording has been changed as suggested. (line 104)

Line 108: Please add reference.

Line 110: Please add reference.

Response: References have been added on these lines. (lines 113 and 115)

Discusssion:

Lines 241-243: The authors stated "treadmill velocities used in these soccer game simulations were similar to what was assessed by the global position tracking system in the current study, we can assume the heart rates and substrate oxidation rates would have been similar in the games assessed in the current study". It is possible to compare data obtained during the treadmill test with a soccer match where repeated bouts of intermittent exercise intersped with different actions such as continuous changes-of-direction, accelerations and decelerations are involved? Please explain carefully.

Response: We have added that this comparison is limited because data obtained from a treadmill simulation cannot perfectly match actions of a soccer match where repeated bouts of intermittent exercise are interspaced with different actions such as continuous changes of direction, accelerations and decelerations. (line 265)

Conclusions:

The conclusions should be "A low-glycemic index pulse-based diet was beneficial for improving serum lipid profile in female but not male soccer players; however, the low-glycemic index diet did not have a beneficial effect on performance in the current study compared to the high-glycemic index diet".

Response: We have changed our conclusion statement to that suggested by the reviewer. (line 271)